# Nanopore sequencing of intact aminoacylated tRNAs

Laura K. White [1,5] ✉, Aleksandar Radakovic[2,3,5], Marcin P. Sajek [1,4], Kezia Dobson[1], Kent A. Riemondy[1], Samantha del Pozo [1], Jack W. Szostak [3] & Jay R. Hesselberth [1] ✉

The intricate landscape of tRNA modification presents persistent analytical challenges, which have impeded efforts to simultaneously resolve sequence, modification, and aminoacylation state at the level of individual tRNAs. To address these challenges, we introduce "aa-tRNA-seq", an integrated method that uses chemical ligation to sandwich the amino acid of a charged tRNA in between the body of the tRNA and an adaptor oligonucleotide, followed by high throughput nanopore sequencing. Our approach reveals the identity of the amino acids attached to all tRNAs in a cellular sample, at the single molecule level. We describe machine learning models that enable the accurate identification of amino acid identities based on the unique signal distortions generated by the interactions between the amino acid in the RNA backbone and the nanopore motor protein and reader head. We apply aa-tRNA-seq to characterize the impact of the loss of specific tRNA modification enzymes, confirming the hypomodification-associated instability of specific tRNAs, and identifying additional candidate targets of modification. Our studies lay the groundwork for understanding the efficiency and fidelity of tRNA aminoacylation as a function of tRNA sequence, modification, and environmental conditions.

Transfer RNAs (tRNAs) are the fundamental adaptors of translation, linking mRNA decoding and polypeptide synthesis[1]. Biogenesis of tRNAs is complex: following transcription, precursor tRNA molecules undergo multiple trimming and modification events, yielding small (~76 nt) and densely modified (~13 modifications/molecule) RNA molecules that are then competent for aminoacylation (reviewed in ref. [2]). tRNA modifications can profoundly impact tRNA function through their impacts on stability[3], aminoacylation efficiency[4,5], and expanded decoding capacity[6]. tRNA charging levels vary in response to environmental conditions[7,8] and drive various physiological responses to stress[9], starvation[10,11], and proliferation[12,13]. In addition, aminoacylation of tRNAs with non-cognate amino acids ("misaminoacylation") has

been observed at high rates in aminoacyl synthetases with proof-reading defects and during oxidative stress[14–18].

High throughput analysis of tRNAs has been hampered by a lack of incisive methods to comprehensively and directly measure tRNA abundance, modification, and aminoacylation within the same experiment. Short-read DNA sequencing approaches have been developed to study tRNA charging, but begin by selective oxidation and β-elimination of non-aminoacylated tRNAs, marking this population through removal of the 3'-terminal nucleotide, followed by dea-cylation, adapter ligation, and reverse transcription[19–21]. However, the large number of modifications installed on tRNAs make them poor substrates for reverse transcriptase (RT) enzymes. In addition, most

[1]Department of Biochemistry and Molecular Genetics, University of Colorado School of Medicine, Aurora, CO, USA. [2]Department of Genetics, Harvard Medical School, Boston, MA, USA. [3]Department of Chemistry, Howard Hughes Medical Institute, The University of Chicago, Chicago, IL, USA. [4]Polish Academy of Sciences, Institute of Human Genetics, Poznan, Poland. [5]These authors contributed equally: Laura K. White, Aleksandar Radakovic. ✉e-mail: laura.k.white@cuanschutz.edu; jay.hesselberth@cuanschutz.edu

tRNA modifications do not produce detectable RT signatures[22] or require chemical derivatization to do so[23]. As such, current sequencing approaches for studying aminoacylated tRNAs rely on indirect information to analyze key properties of these molecules, using chemical and enzymatic treatments to infer the positions of body modifications and presence of charged amino acids. Hence, lower throughput and more cumbersome techniques capable of physically separating uncharged tRNA and aminoacylated tRNA (aa-tRNA)[24,25] remain the gold standard for quantification of tRNA aminoacylation.

A promising new direction is the application of nanopores to tRNA sequencing. Successful discrimination of several tRNAs by solid-state[26] and biological[27,28] nanopores has been followed by methods to directly sequence tRNAs on a commercially available nanopore platform[29], enabling direct analysis of the RNA molecule[30]. Further refinement in adapter design and computational analysis have improved tRNA nanopore sequencing to the point that it is now possible to directly assess tRNA abundance and modification status with single molecule precision[31,32].

Here, we develop a nanopore sequencing approach ("aa-tRNA-seq") that directly captures information on tRNA sequence, modifications, and aminoacylation in a single read. Our method enables selective capture of tRNAs based on their aminoacylation status, selectively embedding the amino acid of aa-tRNAs within the adapter-ligated tRNA molecule. In a separate step, we capture non-aminoacylated tRNA, facilitating comparative analyses of tRNA charging. We characterize nanopore signals produced by 20 proteinogenic amino acids using synthetic tRNA, and leverage these signals to train a recurrent neural network (RNN) to discriminate aminoacylated tRNAs from their uncharged counterparts, and extend this approach for pairwise amino acid classification. We apply the method to study changes in budding yeast tRNA populations during nutrient limitation and in conditions of tRNA hypomodification, confirming known and identifying unexpected changes in tRNA aminoacylation and abundance during these stress conditions.

## Results

### A chemical ligation approach enables selective capture of intact aminoacylated tRNAs

While investigating prebiotic roles of aminoacyl-RNAs, we developed a splinted ligation reaction that generates amino acid-bridged chimeric RNA molecules using a 5′-phosphorimidazole-activated oligoribonucleotide[33,34]. We realized that aminoacyl-tRNAs were analogous to these substrates, so we tested and found that a synthetic tRNA-Gly-GCC aminoacylated with glycine or lysine using the Flexizyme[35] underwent chemical ligation with moderate kinetics, while the non-aminoacylated tRNA yielded no detectable product (Supplementary Fig. 1a). To accelerate the reaction between the α-amino group of the aminoacyl-tRNA and the activated adapter, we included 1-(2-Hydroxyethyl)imidazole (HEI) as an organocatalyst[36] (Fig. 1A and Supplementary Fig. 1c), and subjected *Saccharomyces cerevisiae* tRNA to a 30 min chemical ligation at pH 5.5. Under these conditions, we achieved efficient ligation of aminoacylated budding yeast tRNA as measured by acid northern blot, with <0.1% background ligation to a chemically deacylated tRNA control for tRNA-Gly-GCC (Fig. 1B). To examine the efficiency of this reaction on budding yeast tRNA substrates, we stripped and reprobed the same membrane from Fig. 1B using oligonucleotide probes complementary to isodecoders from each of the 20 tRNA isoacceptor families in yeast (Supplementary Fig. 2). Of the 16 tRNA isodecoders that were separated sufficiently to enable densitometric quantification, the percent of aminoacylated species shifted upon chemical ligation ranged from 62 to 100%, with quantitative ligation of arginyl, asparaginyl, cysteinyl, glutaminyl, glycyl, and lysyl tRNA species (Supplementary Fig. 1b). Using a fluorescently labeled tRNA analog, we confirmed this HEI-catalyzed reaction yields nearly quantitative ligation in vitro to substrates that were

Flexizyme-charged with 13 different amino acids (Supplementary Fig. 1b, d).

### A chemical-northern blot simplifies analysis of aminoacylated tRNA

The analysis of tRNA aminoacylation by acidic northern blotting requires careful handling to preserve the labile ester linkage, using a large format, low pH polyacrylamide gel run in cold, acidic buffer conditions to provide adequate resolution of charged and uncharged tRNA, which is achieved by >12 h of electrophoresis depending on the isodecoder of interest[24], though we found some aa-tRNAs (e.g., Glu-UUC, Asp-GUC) are poorly resolved (Supplementary Fig. 2). Chemical ligation of aminoacyl-tRNA stabilizes the ester linkage[33,37,38] and significantly increases its size, enabling robust separation from non-acylated tRNA by acidic northern blot (Fig. 1B). These features motivated a simpler approach to separate charged and uncharged tRNA using non-acidic denaturing polyacrylamide gel electrophoresis. In this "chemical-charging northern" (Fig. 1C), budding yeast tRNA was chemically ligated as in Fig. 1B, followed by a ~30 min electrophoretic separation on a standard TBE-urea mini-gel, membrane transfer, and probe hybridization for the same glycyl isodecoder. We found that 35% of Gly-GCC tRNA displays a gel shift after chemical ligation (Fig. 1C, lane 4, consistent with the value from Fig. 1B), with 7% background ligation to the deacylated sample (lane 2) due to incomplete deacylation (Supplementary Fig. 1e, f[39]). In the absence of the phosphorimidazole-activated 3′ oligoribonucleotide (lanes 1 & 3), no gel shift for the Gly-GCC tRNA was apparent (Fig. 1C).

This strategy of stabilizing the aminoacyl ester and appending a bulky group to the alpha-amine to make it amenable to standard PAGE is reminiscent of earlier studies that leveraged highly-reactive biotin-NHS reagents to biotinylate the alpha-amine, which after incubation with streptavidin, could be resolved from deacylated tRNA with standard PAGE[40-42]. However, our approach offers two key advantages: (1) very low background reactivity with deacylated tRNA (Fig. 1B) due to the low concentration of the activated adapter used in the reaction and (2) direct modification of aminoacylated tRNA with an oligonucleotide adapter, which allows both the resolution of the aminoacylated from the deacylated tRNA by standard PAGE and the subsequent preparation of aminoacylated tRNA libraries for nanopore sequencing (vide infra).

### Aminoacylated tRNAs can be analyzed by nanopore sequencing

We next assessed whether synthetic tRNA ligated to an activated 3′ oligoribonucleotide via a bridging amino acid was amenable to nanopore sequencing. We confirmed that adapters for direct tRNA nanopore sequencing[32] can be attached to budding yeast tRNA via sequential chemical and enzymatic ligations of the 3′ and 5′ sequences, respectively (Fig. 1D), suggesting a clear strategy for the preparation of nanopore sequencing libraries containing aminoacylated tRNAs (Fig. 1E). We synthesized Gly-tRNA charged with the Flexizyme[35], chemically ligated this to the same phosphorimidazolated 3′ adapter, gel-purified the ligation product, and enzymatically ligated the 5′ adapter using T4 RNA ligase 2 (RNL2). We then prepared a nanopore direct RNA sequencing library from the individual ligated Gly-tRNA using the RNA004 chemistry from Oxford Nanopore Technologies (ONT). Figure 1F shows the difference in ionic current between a synthetic tRNA and the same sequence charged with glycine, with lower current for Gly-tRNA spanning multiple nucleotides as the aminoacylated molecule is pulled into the nanopore from its 3′ end during motor-catalyzed translocation.

We next synthesized tRNA charged with each of the 20 standard proteinogenic amino acids using the Flexizyme and subjected them to the same library preparation. Quality control assessment revealed no major issues for 18 of 20 of these libraries; however, 41.0% of Asn-tRNA reads and 27.6% of cysteinyl-tRNA reads terminated aberrantly,

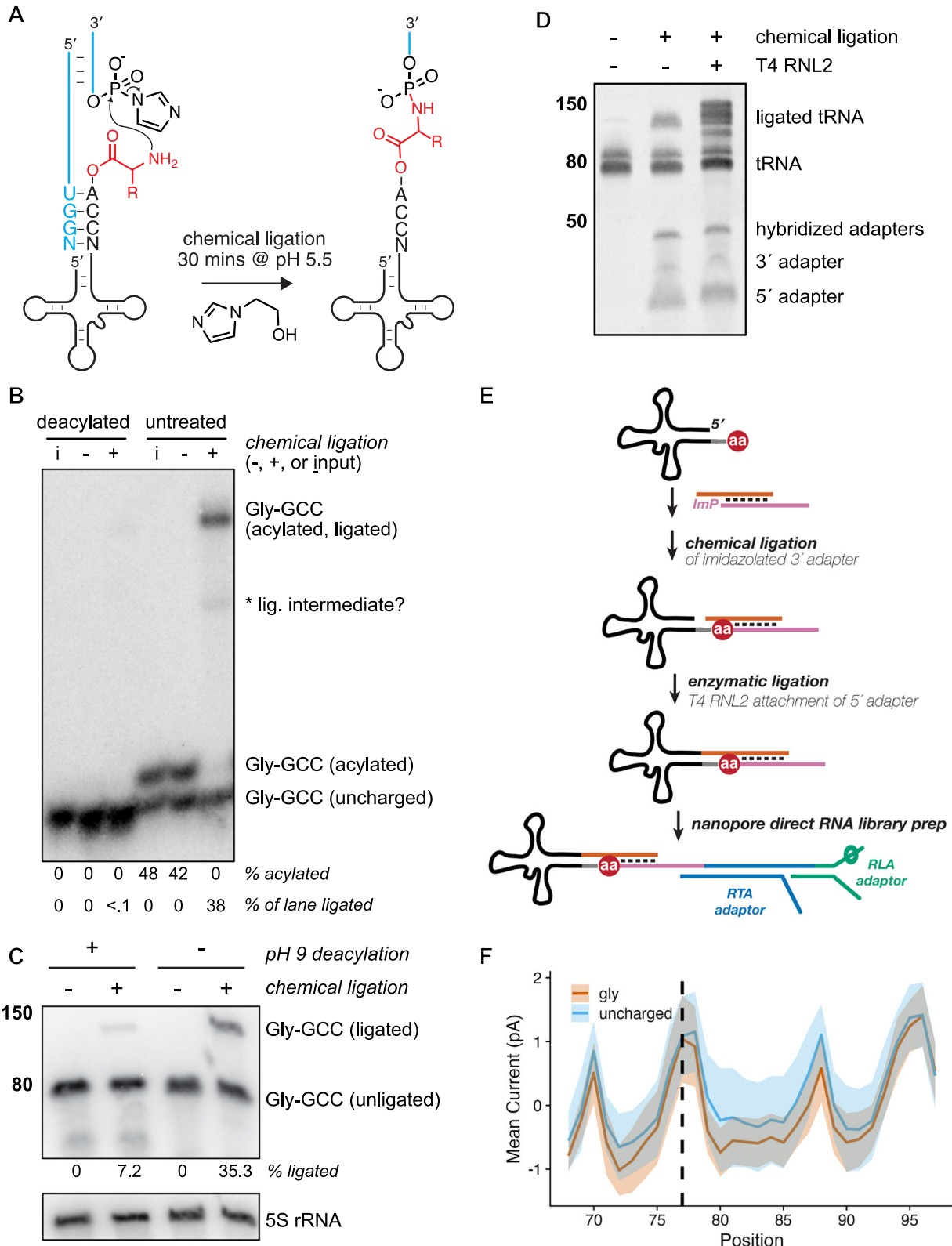

consistent with these tRNAs becoming stuck in nanopores and subsequently ejected ("unblocked") at higher rates than those charged with other, bulkier amino acids or the uncharged tRNA control (Supplementary Fig. 3a). As the Asn-tRNA and Cys-tRNA libraries also had lower sequencing yields, we sought to resolve this issue and tested multiple strategies, illustrated in Supplementary Fig. 3b. For Cys-tRNA, either (i) alkylation with chloroacetamide after chemical ligation or (ii)

omitting the enzymatic ligation to the 5′ adapter resolved the pore blocking (Supplementary Fig. 3c). This second strategy also reduced pore blocking for tRNA-Asn, suggesting that the thiol of Cys and amide of Asn might interact with the 5′ RNA adapter, leading to the molecule becoming stuck in the nanopore during sequencing. Further testing revealed that changing the 5′ adapter from an all-RNA oligonucleotide to a DNA oligonucleotide with five ribonucleotides at its 3′ terminus

**Fig. 1 | A chemical ligation strategy for capture of aminoacylated tRNAs.**
**A** Schematic of splinted chemical ligation of 5′-phosphorimidazolide activated adapter (blue) to aminoacyl-tRNA (black, with amino acid in red) in the presence of the catalyst 1-(2-Hydroxyethyl)imidazole (HEI). **B** Acidic charging northern of deacylated or untreated wild-type (BY4741) *S. cerevisiae* tRNA after chemical reaction in the presence (+) or absence (−) of activated 3′ adapter, compared to tRNA input only (I). Blot has been probed for budding yeast tRNA-Gly-GCC; asterisk indicates a presumed ligation intermediate. Experiment was performed independently twice with similar results. **C** "Chemical-charging northern blot" resolving charged and uncharged tRNA-Gly-GCC via chemical ligation and analysis on a mini (8 × 10 cm) TBE/7 M urea polyacrylamide gel followed by membrane transfer and hybridization with a Gly-GCC probe. Band sizes correspond to a single-stranded RNA ladder (in nucleotides, nt). Experiment was repeated multiple times during method development with consistent results. **D** Small RNAs (17–200 nt, 200 ng) from budding

yeast were chemically ligated to activated 3′ RNA adapter in a splinted ligation as previous panels, followed by an optional enzymatic ligation with T4 RNA ligase 2 (RNL2) to attach a second, splinted RNA adapter to the 5′ end of the tRNA. Products were run on a 10% denaturing PAGE gel and stained with SYBR Gold. Band sizes correspond to a single-stranded RNA ladder (in nucleotides, nt). Source data for (**B**–**D**) are provided as a Source Data file. **E** Schematic illustrating strategy for chemical ligation and sequencing of charged and uncharged biological tRNAs via nanopore direct RNA sequencing. **F** Normalized mean current in picoamps for synthetic tRNA charged with glycine vs an uncharged control. The region visualized includes the 3′ terminus of the tRNA (6 nt, positions 68–73), the CCA tail (positions 74–76), the aminoacylated position (dashed line), and the entirety of the 3′ adapter sequence. For each colored trace, the solid line is the mean signal, and the shading spans the standard deviation.Source Data.

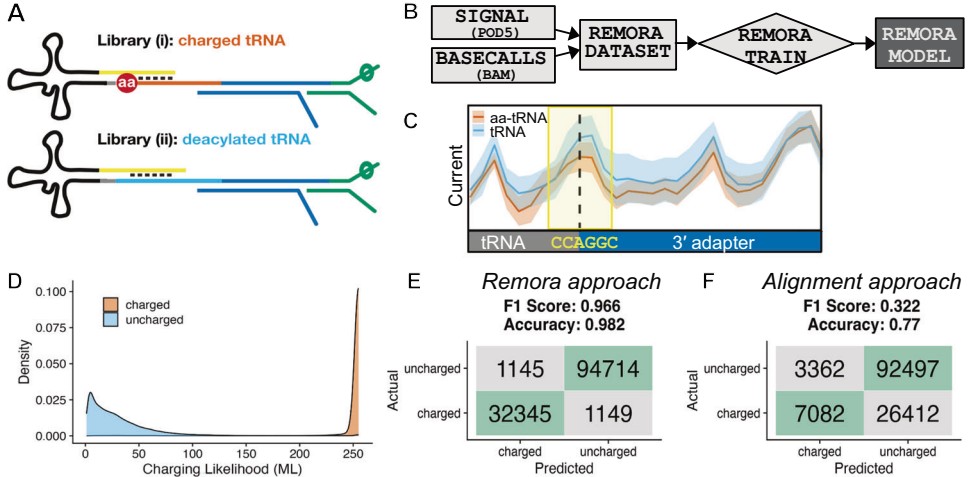

**Fig. 2 | Model training and classification of aminoacylated tRNA reads from nanopore signal.** **A** Schematic illustrating ground truth libraries for model training. Library *(i)* was prepared by only chemical ligation of the 3′ adapter to biological aminoacylated tRNAs, followed by gel purification and enzymatic ligation of the 5′ adapter. Library *(ii)* was prepared by enzymatic ligation of both adapters to deacylated biological tRNAs. **B** Strategy for dataset preparation and neural network training of a Remora model to identify charged and uncharged tRNA reads from nanopore current signals. **C** Schematic depicting reference-anchored signal from libraries *(i)* and *(ii)*, with the 6-nt training window enclosed within the yellow box,

and the position of the amino acid via dashed line. Mean current signals in picoamps for charged and uncharged tRNA reads are indicated by solid lines within each colored trace, with corresponding shading spanning the standard deviation. **D** Density plot showing the distribution of aminoacylation prediction scores (modification likelihood or "ML" score, 0–255 scale) assigned by Remora model in validation libraries (95,859 deacylated tRNA reads, 33,494 aminoacylated tRNA reads). **E** Confusion matrix and key metrics illustrating the model's performance at classifying a second set of replicates prepared as in (**A**), using a cutoff of ML > 199 for identification of aminoacylated reads from nanopore signal.

(i.e., nearest the amino acid) ameliorated the issue for both Cys and Asn (Supplementary Fig. 3c, d); therefore, this DNA/RNA hybrid adapter was used in all subsequent experiments.

## Discrimination of charged and uncharged biological tRNAs by nanopore sequencing

Modified basecalling of nanopore sequencing signals is achieved by training neural networks[43]; in the case of direct RNA sequencing, detecting the presence of a modification via comparison to an unmodified control is often a more trivial computational challenge than precise characterization of a modification's identity[44–47]. To extend this concept to aminoacylated tRNA sequencing, we sought to classify charged and uncharged tRNA sequencing reads, with the goal of developing a one-pot approach for nanopore tRNA sequencing to capture charged and uncharged molecules within the same sequencing library. We prepared separate "ground truth" libraries containing either (*i*) chemically ligated, charged tRNA, or (*ii*) deacylated, enzymatically ligated tRNA from budding yeast (Fig. 2A). tRNAs were ligated to 3′ adapters containing unique sequences for charged and uncharged molecules, and a common 5′ adapter incorporating the design changes (DNA/RNA hybrid) from Supplementary Fig. 3b, and sequenced these libraries on PromethION *RNA* flow cells.

Informed by our synthetic tRNA sequencing results (Fig. 1F), we trained a RNN on this dataset using the Remora software from ONT (Fig. 2B), which predicts the presence of modified nucleotides using ionic current signals generated during nanopore sequencing. For training, we defined a six-nucleotide signal window spanning the invariant CCA at the 3′ terminus of mature tRNA ("CCA-G-GC", where "G" represents the first base of the adapter most affected by the amino acid, and "GC" represent the next two bases of the adapter; depicted in Fig. 2C), as the sequence in this region would be shared across all tRNA isodecoders. We generated a Remora dataset from 80% of these reads, with labels defined by the library of origin (*i* and *ii*, above), and then trained and evaluated a model on the reserved 20% test set. During signal-anchored inference, Remora models output predictions for "modified" positions in the ML ("modification likelihood") tag of a BAM file; these values range from 0 to 255, where lower values are more likely to be canonical nucleotides. Inspection of these values in our libraries revealed a bimodal distribution, with 96.6% of reads from the charged library (*i*) bearing ML scores ≥200, compared to 1.2% of in the deacylated sample (*ii*) (Fig. 2D).

We generated a second biological replicate of the libraries in Fig. 2A for model validation, performing reference-anchored inference using the trained model above and using a ML cutoff of >199 to classify

aminoacylated tRNAs. Figure 2E illustrates the Remora model's performance on this new dataset, with an F1 score of 0.966. This signal-based approach substantially outperformed an alignment-based one using unique 3′ adapter sequences to discriminate charged and uncharged tRNAs, which yielded an F1 score of 0.322 with low sensitivity for identifying charged tRNA reads (Fig. 2F), and which we attribute to increased base-calling error caused by the embedded amino acid. Based on its clear improvement over the alignment-based approach, we focused on signal-based classification of tRNA aminoacylation state.

### Measurement of tRNA aminoacylation with aa-tRNA-seq

To determine whether this approach could report on tRNA charging levels, we generated replicate aa-tRNA-seq libraries from wild-type budding yeast under conditions identical to Fig. 1B (wild-type yeast in rich media), enabling an orthogonal comparison of tRNA aminoacylation measured by acidic northern (Fig. 1B) and model-based classification of charged and uncharged nanopore sequencing reads. Across the 16 tRNA isodecoders quantifiable by densitometry (Supplementary Fig. 1b), we observed a Pearson's correlation of 0.68; however, estimates of isodecoder charging by sequencing were uniformly lower than charging levels measured by northern (Fig. 3A). While in principle this difference might be explained by differences in the rates of aberrant read termination between charged and uncharged molecules, the magnitude of this result was inconsistent with our observations in synthetic tRNA sequencing data after optimization of the 5′ adapter sequence (Supplementary Fig. 3). We therefore re-analyzed the initial dataset used for Remora model training, comparing the translocation time for tRNA reads in the first library prepared via chemical ligation to aminoacylated tRNA to the second library where tRNAs were deacylated and enzymatically ligated. This analysis revealed that the total translocation time for aminoacylated tRNA—across all isodecoders—is on average ~1.2-fold longer than their uncharged counterparts (Supplementary Fig. 4), which is likely the primary driver of the underestimation of charging we observe in aa-tRNA-seq libraries (Fig. 3A): in a complex mixture of biological tRNAs, uncharged tRNAs are sequenced faster than charged tRNAs, leading to a sampling bias toward uncharged tRNAs. As such, while absolute quantitation of aa-tRNAs requires further optimization, we focused on relative quantitation of tRNA abundance and charging.

### aa-tRNA-seq detects changes in tRNA aminoacylation during nutrient stress

We next sought to assess whether aa-tRNA-seq could detect dynamic changes in tRNA charging and focused on nutrient limitation, which causes rapid changes to the aminoacyl-tRNA pool. We subjected a *S. cerevisiae* leucine auxotroph to leucine starvation for 15 min, and found that, as previously reported[11], this caused rapid depletion of aminoacylated leucyl-tRNA isodecoders (Fig. 3B–D). Using a chemical-charging northern (Fig. 3B), we observed a mean 28.4% decrease in the abundance of aminoacylated leucyl-tRNA. We performed aa-tRNA-seq on the same material, and examined global changes in tRNA abundance and aminoacylation upon leucine starvation. This analysis revealed a significant decrease in the charging of all four leucyl tRNA isodecoders present in budding yeast, while most other tRNAs remained unaffected. Surprisingly, we also found a concomitant increase in the charging of Ala-tRNA isodecoders (3.97-fold for Ala-UGC, *p* < 0.001; 2.12-fold for Ala-AGC, *p* = 0.08; BH-adjusted *Z* test; Fig. 3C, D), an effect that was not detected in earlier microarray-based experiments examining tRNA charging in response to leucine starvation[11]. We validated this isodecoder-specific effect by chemical-charging northern, where we observed a mean 2.62-fold increase for Ala-TGC compared to a mean 1.10-fold increase across three other unaffected isodecoders (Supplementary Fig. 5).

### aa-tRNA-seq detects interdependence between tRNA modifications and aminoacylation during rapid tRNA decay

Hypomodified tRNAs are susceptible to "rapid tRNA decay" (RTD) in budding yeast, fission yeast, and bacteria[48–56]. For example, Val-AAC tRNA lacking 5-methyl cytidine and 7-methyl guanine in *trm8Δ trm4Δ* budding yeast is rapidly destabilized and degraded at high temperature by 5′–3′ exonucleolytic decay[57]. We cultured a *trm4Δ trm8Δ* strain and a control bearing an additional disruption of *MET22* at the permissive temperature of 28 °C followed by a shift to the non-permissive temperature of 37 °C for three hours. Deletion of *MET22* causes accumulation of pAp (adenosine 3′,5′ bisphosphate, a competitive inhibitor of 5′–3′ exonucleases[58]), suppressing RTD[57]. We isolated small RNA from each strain and temperature in biological triplicate, and prepared this material for analysis by chemical-charging northern and aa-tRNA-seq. Both approaches confirmed defects in stability and aminoacylation for Val-AAC[57]. By chemical-charging northern, aminoacylation levels for Val-AAC drop ~twofold upon shift to the nonpermissive temperature in *trm8Δ trm4Δ* cells, and this effect was suppressed by *met22Δ* (Fig. 3E). This effect is readily detected by aa-tRNA-seq, where Val-AAC undergoes significant changes in both tRNA abundance (-3.6-fold reduction, *p* = 2.93 × 10⁻⁸, BH-adjusted Z test) and aminoacylation (-3.3-fold decrease, *p* = 1.99 × 10⁻⁶, BH-adjusted *Z* test) upon temperature shift (Fig. 3F). Consistent with previous reports[48], in yeast strains bearing single deletions of *TRM4* or *TRM8*, Val-AAC aminoacylation remains near constant upon shift to the non-permissive temperature (Supplementary Fig. 6d, e).

We also observed statistically significant increases in aminoacylation (2.54-fold for Gly-GCC, *p*-value = 0.009; 2.73-fold for Gly-CCC, *p*-value = 0.004, BH-adjusted *Z* test) for two glycyl-tRNA isodecoders upon a shift to 37 °C (Fig. 3F, G), which were eliminated by *MET22* co-deletion in our sequencing data (Fig. 3H). However, validation of this result by chemical-charging northern was less conclusive. While we detected a 1.22-fold increase in Gly-GCC aminoacylation in the double mutant, aminoacylated Gly-GCC tRNA in the *trm8Δ trm4Δ met22Δ* triple mutant also increased by a similar magnitude (Supplementary Fig. 6a). Two additional isodecoders, Cys-GCA and Leu-CAA, also displayed more modest (~1.1-fold increases) changes in aminoacylation upon temperature shift (Supplementary Fig. 6b, c). To investigate this result further, we again turned to single deletions of *TRM4*, *TRM8*, and *MET22*. For Gly-GCC, the mean change in aminoacylation upon temperature shift was 1.54-fold for *trm4Δ*, 1.20-fold for *trm8Δ*, and 1.27-fold for *met22Δ*, respectively (*p*-value of 0.09 by ANOVA, Supplementary Fig. 6f, g). While not statistically significant across biological replicates, the results hint at a potential distinct role for *TRM4* in modulating Gly-GCC aminoacylation during temperature stress, in contrast to *TRM8*, which is not known to act directly on Gly-GCC[59].

RTD has also been described in *tan1Δ trm44Δ* cells, which are temperature sensitive due to lack of acetylation at $C_{12}$ by *TAN1* and loss of methylation at $U_{44}$ by *TRM44*; these enzymes share activity on multiple serine and leucine isodecoders in budding yeast[57,59]. As in *trm8Δ trm4Δ*, temperature sensitivity is accompanied by exonuclease-mediated decay of specific tRNA isodecoders (Ser-UGA, Ser-CGA) and both the molecular and growth phenotypes are alleviated by co-deletion of *MET22*[57,60]. We grew *tan1Δ trm44Δ* and *tan1Δ trm44Δ met22Δ* cells under the same RTD-inducing conditions described above, isolated small RNA, and again performed chemical ligation and analysis by chemical-charging northern and aa-tRNA-seq. Our results indicate that the targets of RTD in *tan1Δ trm44Δ* cells are restricted to the tRNA species originally identified; however, aminoacylation of these hypomodified tRNAs is unaffected upon temperature stress. After three hours of temperature stress, we detected modest decreases in abundance of both serine tRNA isodecoders previously reported to be affected in *tan1Δ trm44Δ* cells via aa-tRNA-seq, with levels of Ser-CGA (the more strongly affected isodecoder in previous studies)

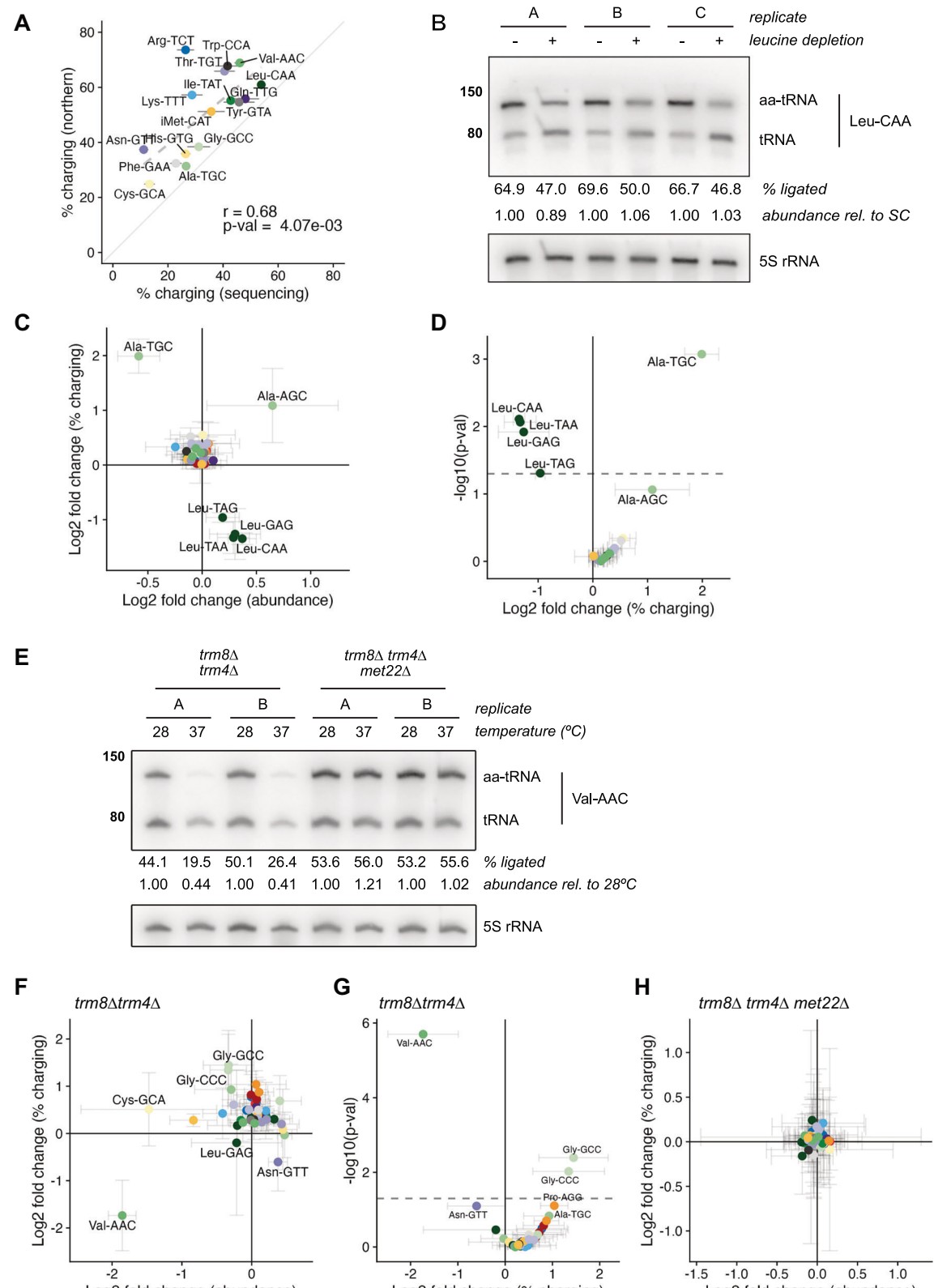

decreasing by a statistically significant 22.2% and Ser-UGA showing a 13.8% reduction in abundance upon temperature stress (Supplementary Fig. 7a–c). When validated by chemical-charging northern, *tan1Δ trm44Δ* cells displayed 6–11% drops in Ser-UGA abundance upon shift to the nonpermissive temperature (Supplementary Fig. 8a–c), while levels of the unaffected isodecoder Leu-CAA remained constant or increased.

## A machine learning approach to discriminate embedded amino acids in aa-tRNA-seq

Because the identity of the amino acid on an aminoacylated tRNA is invisible to other tRNA sequencing approaches, a major advance for the field would be the detection of misaminoacylation events directly from high throughput tRNA sequencing libraries. Towards this goal, we performed detailed analysis of signals produced during nanopore

**Fig. 3 | Sequencing and analysis of budding yeast tRNAs via chemical-charging northern and aa-tRNA-seq. A** Correlation between tRNA aminoacylation (% charging) as measured by acid northern (*y*-axis), and aa-tRNA-seq (mean ± SD of 3 biological replicates). Dashed line indicates Pearson correlation (two-sided; *r* and *p*-value shown); solid line indicates *y* = *x*. **B** Chemical-charging northern of Leu-CAA aminoacylation in a leucine auxotroph after 15 m leucine depletion. Shown are 3 biological replicates; ligated tRNA (% per lane) was normalized to 5S rRNA (lower inset). Calculated abundance values were normalized within each replicate to the sample grown in complete media (set to 1.0) to assess relative abundance under leucine-starved conditions. Source data are provided as a Source Data file. **C** Log$_2$ fold change in tRNA abundance and percent charged reads for all isodecoders from the same 3 biological replicates as (**B**). Data are shown as mean ± SD. **D** Volcano plot of mean aminoacylation fold change between leucine-depleted and complete media, based on replicates in (**B**, **C**). Each point represents the mean value for a tRNA isodecoder. *P*-values from two-sided *Z*-test with Benjamini–Hochberg correction; dashed line indicates the α threshold. **E** Chemical-charging northern Val-

AAC tRNA in *trm8Δ trm4Δ* (RTD-sensitive) and *trm8Δ trm4Δ met22Δ* (RTD-resistant) strains after 3 h at 28 °C or 37 °C. Two biological replicates shown; signals normalized to 5S rRNA. The percent of chemically ligated Val-AAC tRNA (upper band) is indicated below each lane. Val-AAC tRNA abundance levels were normalized within each replicate to the sample grown at 28 °C (set to 1.0) and compared to a matched sample shifted to 37 °C. Source data are provided as a Source Data file. **F** Log$_2$ fold change in tRNA abundance and percent charged reads in *trm8Δ trm4Δ* cells after temperature shift, based on 3 biological replicates, including those in (**E**). Data shown as mean ± SD. **G** Volcano plot of mean fold change in aminoacylation between 37 °C and 28 °C for *trm8Δ trm4Δ* cells, using same replicates as (**E**, **F**). *P*-values from two-sided *Z*-test with Benjamini–Hochberg correction; dashed line indicates the α threshold. Data are presented as mean values. **H** Log$_2$ fold change in tRNA abundance and percent charged reads upon temperature shift in *trm8Δ trm4Δ met22Δ* cells, based on *n* = 3 biological replicates. Data are shown as mean ± SD.Source Data.

sequencing for aminoacyl-tRNA charged with each of the 20 proteinogenic amino acids using the Flexizyme system. While nanopore basecallers and other machine learning approaches for detection of modified bases are typically trained on ionic current signatures produced at specific residues[47,61,62], the raw signal generated during nanopore sequencing is a composite of changes in current and translocation speed (measured in "dwell time", the time between inferred translocation states). Because they are sequence-identical (with the exception of Cys- and Asn-tRNA libraries, which were prepared using the DNA/RNA 5′ adapter sequence tested in Supplementary Fig. 4 and used in all biological sequencing experiments), our synthetic aa-tRNA-seq libraries enable the isolation of amino acid specific signals by comparing each of the 20 aa-tRNAs to an uncharged control.

Aminoacylated tRNAs yield specific distortions in nanopore signals, generating unique signatures that vary between amino acids. Figure 4A displays the mean dwell time in milliseconds for each of our 20 synthetic aa-tRNA libraries and an uncharged control tRNA. We observed large increases in dwell time for charged tRNAs at a position located 9 nucleotides downstream from the amino acid (position 86), with dwell times exceeding 1 s for 9 of 20 amino acids. Because nanopore direct RNA sequencing proceeds in a 3′ to 5′ direction, we speculate that this signal represents specific interactions between amino acid and the motor protein as the 3′ adapter is transiting through the nanopore reader head. Figure 4B shows the relative change in normalized current for each of the aminoacylated tRNA reads over this same window, compared to the uncharged tRNA control. Charged tRNA libraries display lower mean current values than the non-aminoacylated substrate across most of this region, suggesting that amino acids occlude ionic flow as they transit through the helicase/pore assembly. Consistent with this explanation, we observed the largest reductions in mean current at the precise site of aminoacylation for the most bulky amino acids, indicating that the largest distortions in ionic current distortions occur within the narrowest aperture of the nanopore itself.

We examined the correlation between the largest current and dwell time effects and various amino acid properties. While the putative helicase interactions at position 86 are poorly correlated with amino acid volume and molecular weight (Fig. 4C, D), we identified strong correlations between dwell time and hydrophobicity (Fig. 4E), as well as correlations between amino acid mass and volume, and the observed current differences at position 77 (the position of the amino acid; Fig. 4F, G). Together, these observations likely reflect straightforward physical occlusion of ion flow within the nanopore reader head and more complex interactions between the tRNA-embedded amino acid and the motor protein.

Closer examination of our data revealed substantial variation in the magnitude of current (Supplementary Fig. 9) and dwell time (Supplementary Fig. 10) signals produced by different synthetic

aminoacyl-tRNAs. We trained 380 pairwise models to discriminate one AA from another (Fig. 4I), and evaluated their performance on our synthetic data (Fig. 4J). The performance of these models was generally high with a median F1 score of 0.86, indicating that differences in nanopore current between several pairs of amino acids represent a strong signal for classification. Notable underperforming outliers included isoleucyl-tRNAs, which are poorly distinguished from other hydrophobic side chains, and other specific comparisons (e.g., Phe-Tyr, Pro-Ala, Cys-Arg). We are currently exploring the improvement of these models by explicit inclusion of dwell time information during model training, and plan to apply these models to understand specific cases of potential tRNA misaminoacylation (Fig. 3D, G).

## Discussion

Transfer RNAs are long-lived and undergo repeated cycles of charging and deacylation throughout their lifetime. While tRNA biogenesis is recognized as an intricately coordinated process—spanning transcription, processing, modification, and aminoacylation—less is known about the molecular transformations that tRNAs undergo throughout their functional lifetime and the specific events that trigger their turnover. We developed a chemical ligation approach to selectively capture aminoacylated tRNAs and used it to study tRNA aminoacylation in a variety of contexts. Increased accuracy of aa-tRNA identification may further enhance our understanding of how tRNA modifications act as determinants or anti-determinants of synthetase recognition and specificity. Moreover, by enabling simultaneous screening of tRNA modifications, charging, and stability, this approach provides a versatile framework for engineering synthetic or therapeutic tRNAs with precisely tuned properties[63–67]. Direct identification of amino acids and their corresponding tRNA sequences could also advance studies on aminoacyl-tRNA synthetase evolution and engineering, bypassing the indirect readouts and negative selections commonly employed by current synthetase engineering approaches[41,42,68,69].

We leveraged chemical ligation to simplify the analysis of aminoacylated tRNAs via northern blotting (Fig. 1C). This approach offers key advantages, including simplified electrophoretic separation of chemically ligated species, and is therefore a compelling alternative to more laborious techniques (but see refs. 40–42). However, it is important to note that under the existing reaction conditions, there is some variability in ligation efficiency among biological tRNA isoacceptors and only some were ligated quantitatively (Supplementary Fig. 1b), limiting the performance of this method compared to a traditional acidic northern blot. It is also noteworthy that chemical ligation is likely useful beyond the capture of aminoacylated tRNA: a variety of nucleophiles react with phosphorimidazole-activated oligonucleotides, including RNA terminal hydroxyl groups at elevated pH[70]. Juxtaposition of the phosphorimidazolide near the α-amino

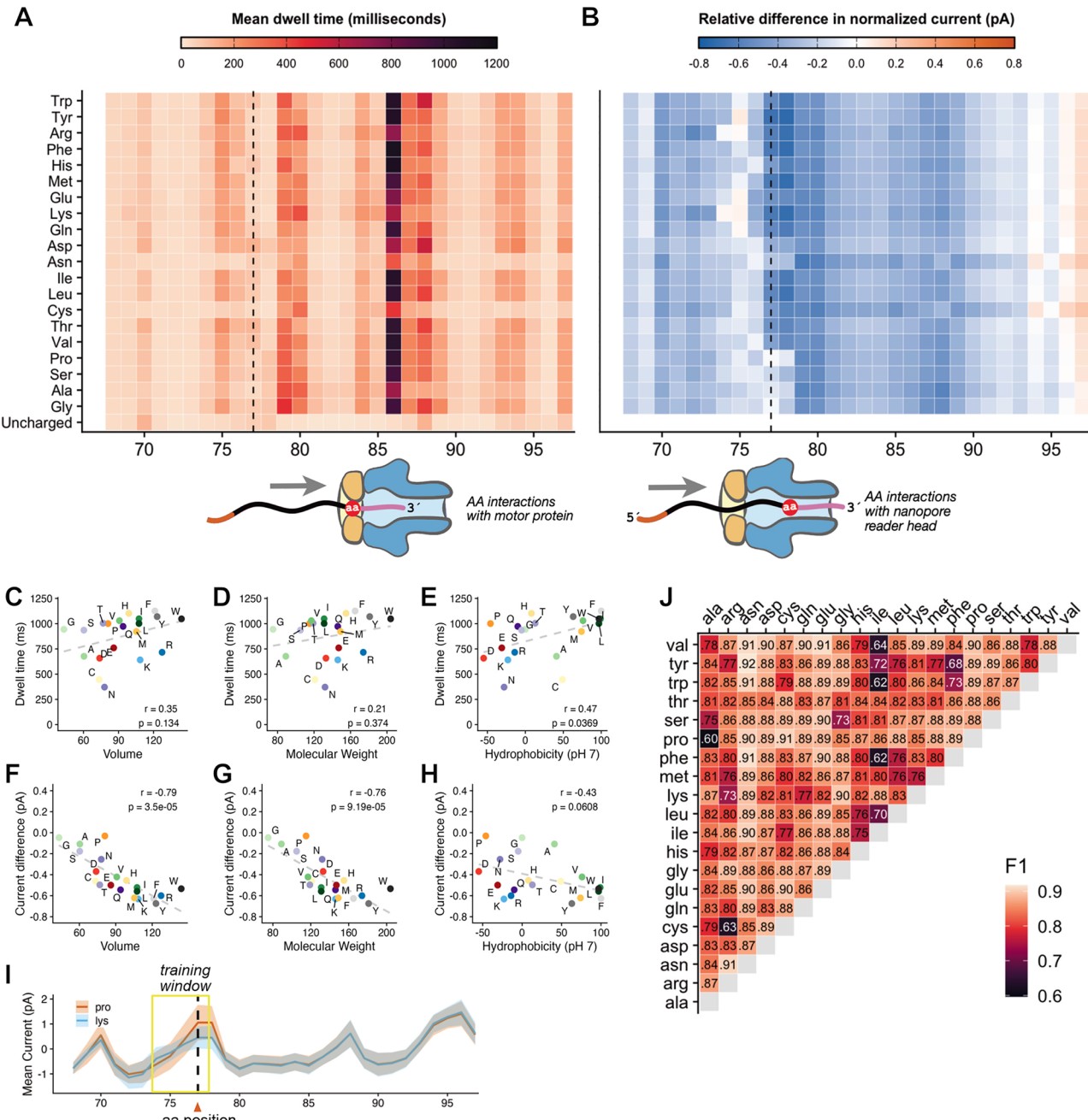

**Fig. 4 | Signal analysis and classification of amino acid identity using nanopore sequencing. A** Mean dwell time for synthetic tRNAs charged with each of the 20 amino acids and an uncharged control, sequenced after enzymatic ligation to the 3′ adapter. Sequences are ordered top to bottom on the *y*-axis by amino acid side chain molecular weight. The plotted region spans the tRNA 3′ terminus (positions 68–73), the CCA tail (74–76), the aminoacylated position (77, dashed line) and complete 3′ adapter. The schematic below indicates the translocation direction and approximate amino acid position within the helicase motor (orange) when the 3′ adapter sequence is centered in the nanopore (blue). **B** Relative differences in normalized mean current between charged and uncharged synthetic tRNAs across the same region as (**A**). Schematic shows approximate amino acid location within the nanopore during maximal current change. **C** Scatter plot showing the relationship between mean dwell time at position 86 and amino acid side-chain volume (in cubic angstroms); the y-axis shows relative difference in mean current. Each

point represetns one amino acid. Dashed line indicates linear regression fit; *r* and *p*-values are from a two-sided Pearson correlation. **D** As in (**C**), but with molecular weight (g/mol) on the *x*-axis. **E** As in (**C**), but with hydrophobicity index[91] at pH 7 on the *x*-axis. **F** Scatter plot showing the relationship between mean current differences at position 77 and amino acid volume (cubic angstroms). As in (**C**) for all statistical and plotting conventions. **G** As in (**F**), but with molecular weight (g/mol) on the *x*-axis. **H** As in (**F**), but with hydrophobicity index[91] at pH 7 on the *x*-axis. **I** Schematic of pairwise Remora model training using signal distributions from synthetic tRNAs charged with proline vs lysine. Shaded regions represent standard deviation around the mean current; dashed line indicates the amino acid position. A yellow box outlines the 4-nt signal window used for training, with the amino acid location marked by a dashed line. **J** F1 scores for all pairwise amino acid classification models trained on synthetic tRNA reads using the strategy in (**I**).

group of the aa-tRNA was facilitated by splinted base-pairing to the 3′-CCA overhang (Fig. 1A), but one could imagine other sequence-specific targeting scenarios, including the 3′-ends of mRNA poly-A tails.

Next, we adapted chemical ligation to enable direct nanopore sequencing of aminoacylated tRNA ligated to a downstream oligonucleotide ("aa-tRNA-seq"). We trained a Remora classification model to distinguish charged and uncharged tRNAs from nanopore sequencing data, and showed that this signal-based classification approach outperformed an alignment-based one (Fig. 2). Our application of aa-tRNA-seq confirmed known effects of hypomodification and nutrient deprivation on tRNA stability and aminoacylation (Fig. 3), revealing that despite the broad specificity of *TRM8* and *TRM4*, their co-deletion uniquely impacts the stability of Val-AAC (Fig. 3F). Similarly, the combined impact of *TAN1* and *TRM44 deletion* is largely restricted to Ser-CGA and Ser-UGA stability (Supplementary Fig. 7). We also found significantly increased aminoacylation for two isoacceptor families in cases of hypomodification (Gly-GCC and Gly-CCC, Fig. 3G) and nutrient deprivation (Ala-TGC, Fig. 3D). These may simply be increases in cognate aminoacylation of these isodecoders that were not previously observed. Alternatively, they may represent new cases wherein a lack of modification or charging enables misaminoacylation of Gly and Ala isodecoders. Methionine misaminoacylation is a common response to stress[71] wherein the methionyl tRNA synthetase non-specifically charges several tRNA isodecoders with methionine. However, the pattern of misaminoacylation we observe is restricted to a few tRNAs, suggesting an alternative mechanism.

Like other tRNA sequencing methods, aa-tRNA-seq has tradeoffs and technical biases. First, we identified intrinsic differences in the total translocation time between charged and uncharged tRNA during nanopore sequencing (Supplementary Fig. 4), which is a barrier to absolute quantitation of tRNA charging. Second, under these reaction conditions, not all biological tRNA isoacceptors are quantitatively ligated (Supplementary Fig. 1b), reflecting an additional source of bias that may be further optimized. Ligation inefficiency reflects a broader challenge: enzymatic adapter ligation bias has complicated other tRNA sequencing methods, often making comparisons of tRNA abundances within the same sample unreliable[72,73]. While methods relying on periodate oxidation to distinguish charged from uncharged tRNAs have proved valuable to the field, they also face technical challenges, including tRNA damage during periodate treatment[21], variability in the sensitivity of tRNA 3′ termini to oxidation, and inconsistency in the number of nucleotides removed during treatment[74]. Additionally, as periodate-based methods require cDNA generation, they are subject to biases from tRNA modifications that inhibit RT processivity. In protocols where these modifications are instead removed, such methods can introduce additional RNA damage from harsh buffer conditions during enzymatic pretreatment[75–77]. To date, we have not tested cDNA conversion in aa-tRNA-seq libraries; while inclusion of a reverse transcription step in nanopore direct RNA sequencing libraries is known to increase sequencing throughput, this addition could introduce additional biases due to non-uniform RT processivity over different amino acids.

Chemically ligated aa-tRNAs generate discrete signals during nanopore sequencing due to interactions between the embedded amino acid and the motor protein and nanopore architecture (Fig. 4). This unique feature enabled accurate identification of bona fide aa-tRNAs while minimizing false positives: we classified 3.6% of reads from the "charged-only" validation library as uncharged. However, some of these may be true non-acylated tRNA that were enzymatically ligated to hydrolyzed phosphorimidazole-adapter, which has a 5′-phosphate competent for T4 RNL2 ligation. While our current approach demonstrates that different amino acids produce distinct signal properties that enable their discrimination in pairwise comparisons (Fig. 4J), further advances in our machine learning approach may enable the direct, de novo identification of amino acids in biological

samples, unlocking new questions about tRNA charging and misaminoacylation.

We anticipate that incorporation of translocation time information into models for classifying aminoacylated tRNAs will generate additional improvements to aa-tRNA-seq. Dwell time provided useful information in the detection of RNA modifications using the previous direct RNA (RNA002) ONT chemistry, including pseudouridine[78], 2′-O-methyl[79], and 2′-phosphate[31] modifications, but existing approaches for training models on nanopore signal do not leverage this information directly, due to the fact that Remora and other software re-anchor ionic current information onto an aligned sequence. While dwell time information is retained in this process, it is transformed and thereby de-emphasized in model training. Notably, the increased dwell times observed for aminoacylated tRNAs—likely due to unique helicase interactions—are a robust signal in our data. However, these effects are less strongly correlated with amino acid properties than the current differences at the aminoacylated position (Fig. 4C–H), where we have focused our model training.

The translocation rate for aa-tRNAs also impacts the pore blocking effects we described for Cys- and Asn-tRNA (Supplementary Fig. 3), as read ejection ("unblocking") is initiated during nanopore sequencing when a constant signal (indicating a stalled molecule) exceeds a set time threshold. While we resolved pore blocking issues via optimization of the 5′ adapter sequence, we do not fully understand why chemical ligation of these substrates produced these artifacts, or why they were resolved by the substitution of deoxyribonucleotides in our 5′ RNA/DNA splint adapter. While Asn-tRNA yielded significant pore blocking, Gln-tRNA did not (Supplementary Fig. 3a), suggesting an issue beyond simply the presence of an amide side chain. We note that the Asn side is uniquely capable of cyclization rearrangements during intein catalysis, which may contribute to its unique pore blocking phenotype[80].

Transfer RNA modifications are installed in evolutionarily conserved but incompletely understood circuits[81,82]. Described links between tRNA modifications and aminoacylation[4,5] remain sparse and poorly characterized, due in part to the lack of incisive and accessible tools to study these relationships. Comprehensive identification and characterization of circuits linking modification and aminoacylation along with tRNA abundance will require further optimization of aa-tRNA-seq and ongoing work to map the 67+ unique RNA modifications present in the tRNA epitranscriptome[59], which will likely require a combination of nanopore sequencing, mass spectrometry[83,84], and other technologies, with the aim of characterizing the complete collection of tRNA modifications (using lower throughput approaches) and understanding the signals they produce during high throughput, direct RNA sequencing experiments[85].

## Methods

### Preparation and chemical ligation of synthetic aminoacylated tRNAs

**Synthesis.** Oligonucleotides listed in Supplementary Table 2 were either purchased from IDT or synthesized on the K&A H-6 RNA/DNA synthesizer. Phosphoramidite coupling times and the remaining synthesis method parameters were as instructed by the manufacturer (ChemGenes and Glen Research). After solid-phase synthesis, oligonucleotides were cleaved and the nucleobases deprotected as recommended by ChemGenes and Glen Research. The cleaved and deprotected solutions were evaporated using a speed-vac for 2 h followed by overnight lyophilization. The dry material was dissolved in 100 μL DMSO to which 125 μL of TEA. 3HF was added followed by incubation at 65 °C for 2.5 h. The fully deprotected oligonucleotides were precipitated with 0.1 volumes of 5 M ammonium acetate and 5 volumes of cold isopropanol. The precipitated material was dissolved in 5 mM EDTA, 99% v/v formamide and purified by denaturing PAGE. The desired gel bands were visualized by UV shadowing, cut out,

crushed, and soaked in 2 mM EDTA, 5 mM sodium acetate on a rotator overnight. The rotated solutions were filtered through a 5 μm syringe filter after which the filtered solutions were concentrated using Amicon MWCO filters. The concentrated solutions were finally precipitated using 0.1 volumes of 3 M sodium acetate and 5 volumes of ethanol, washed twice with 80% v/v ethanol, and air dried.

The 3,5-dinitrobenzyl esters of amino acids (DBE-aas) were synthesized as described in ref. 35 with the following modifications:

1. Boc protecting groups were removed by dissolving the dry crude DBE-Boc-aa material in 2 mL neat TFA and incubating it at room temperature for 10 min. The TFA was removed under a stream of nitrogen and the deprotected product was washed twice with diethyl ether. The diethyl ether was removed under vacuum and the final DBE-aa product was dissolved in DMSO and used in the aminoacylation assays.
2. Boc-Ser, Boc-Thr, and Boc-Tyr (ChemImpex) were purchased with O-tert-butyl protection on the side chain. The deprotection was performed in 90:10 TFA:triethylsilane for 2 h.
3. Boc-Met (ChemImpex) was used without additional side chain protection, but during the TFA deprotection two side products were observed: oxidation to produce a DBE-Met dimer and tert-butylation of the sulfur. This necessitated reversed-phase purification. RediSep Gold® C18 Reversed Phase Column was used with the 5–90% gradient of solvent B (solvent A = 2 mM TEAB pH 8; solvent B = acetonitrile).
4. Boc-Gln and Boc-Asn were purchased with Xan protection on the side chain amide (ChemImpex). DBE-Asn required reversed-phase purification. RediSep Gold® C18 Reversed Phase Column was used with the 5–90% gradient of solvent B (solvent A = 2 mM TEAB pH 8; solvent B = acetonitrile). DBE-Asn additionally requires immediate use after purification due to the presumed rapid intramolecular attack of the side chain amide onto the activated ester.
5. Boc-Cys was purchased with Trt protection on the side chain thiol (ChemImpex). The deprotection was performed in 90:10 TFA:triethylsilane for 2 h.

**Activation.** The 5′-phosphorimidazolide adapter was generated by incubating a solution containing 200 μM of the 5′-phosphorylated adapter, 100 mM imidazole pH 7, and 100 mM EDC.HCl for 2 h at room temperature. The activated adapter was then precipitated by adding 0.1 volumes of saturated sodium perchlorate in acetone and 3 volumes of cold acetone. The pellet was washed twice with a 1:1 v/v solution of acetone:diethyl ether followed by drying under vacuum. The activated adapter was dissolved in 1 mM imidazole pH 8 and stored at −80 °C until use. The same stock of the activated adapter was used throughout the experiment, but care was taken to thaw the stock immediately prior to the experiment, to store it on ice while using it, and to return it to −80 °C as quickly as possible.

**Aminoacylation.** The aminoacylation reactions containing 100 mM HEPES pH 8, 10 mM MgCl$_2$, 40 μM synthetic tRNA, 36.7 μM dFx Flexizyme, 5 mM DBE-aa (20% v/v DMSO), were incubated on ice for 16 h.

**Chemical ligation.** The ligation reactions were set up by diluting the aminoacylation reactions eightfold so that the final solution contained 5 μM of the aminoacylated tRNA, 50 μM 5′-phosphorimidazolide adapter, 50 μM splint (also called the 5′-adapter below), 5 mM EDTA, and 37.5 mM HEPES pH 8. The reactions were allowed to proceed for 24 h on ice, before being diluted with an equal volume of a solution of 5 mM EDTA, 99% v/v formamide and purified by 16% denaturing PAGE. The ligated products were cut out from the gel, crushed, and soaked in a solution of 2 mM EDTA, 5 mM sodium acetate acidified to pH 5 on a rotator for 3 h at 4 °C. The extracted aa-bridged tRNA products were then filtered using 0.22 μm spin filters, concentrated using Amicon 10 k MWCO filters, and desalted using the Oligo Clean and Concentrator kit (Zymo Research).

**Cys-bridged tRNA alkylation.** After chemical ligation and gel purification, the Cys-bridged tRNA was reduced with DTT for 1 h at room temperature. The reaction contained 1.2 μM Cys-bridged tRNA, 50 mM HEPES pH 8, and 10 mM DTT. After the 1 h incubation, the reduction reaction was diluted 1.33-fold so that the final alkylation solution contained 0.9 μM Cys-bridged tRNA, 37.5 mM HEPES pH 8, 7.5 mM DTT, and 50 mM chloroacetamide. The alkylation reaction was allowed to proceed for 30 min in the dark, after which it was cleaned up using the RNAClean XP beads (Beckman Coulter) according to the manufacturer protocol with the following change: immediately after the addition of the bead suspension to the ligation reaction, isopropanol equal to volume of the reaction+beads was added.

### Nanopore library preparation and sequencing of synthetic aminoacylated tRNAs
The chemically ligated tRNA products from above were enzymatically ligated to the 5′-adapter/splint for 30 min at room temperature. The ligation reactions contained 16 pmol of the chemically ligated tRNA, 80 pmol of the 5′-adapter, 1× NEB T4 RNA Ligase 2 buffer supplemented with 5% PEG 8000, 2 mM ATP, 6.25 mM DTT, 6.25 mM MgCl2, and 0.5 units/μL T4 RNA ligase 2 (10,000 units/mL). The ligated material was purified using the RNAClean XP beads (Beckman Coulter) as above. This material was then prepared for nanopore direct RNA sequencing via RTA ligation, which was performed using tRNA purification specific magnetic beads (BioDynami Cat.# 40054S). The remaining library prep and nanopore sequencing was performed as described below on P2solo sequencing instruments, using MinKNOW version 23.11.7.

### Synthetic tRNA mini-substrate experiments
The acceptor stem mimic oligonucleotide (Supplementary Table 2) was aminoacylated with all 20 amino acids as described above. At the end of the 16 h incubation:

1. 1 μL aliquots were diluted in 9 μL of acidic quenching buffer (10 mM EDTA pH 8.0, 1× bromophenol blue, 100 mM sodium acetate pH 5.0, 150 mM HCl, 75% v/v formamide) and analyzed by 20% acidic denaturing PAGE (acidic gels contained 100 mM sodium acetate pH 5.0 instead of the usual 1× Tris-Borate-EDTA). The acidic gels were run in 100 mM sodium acetate pH 5 at 25 W for 3 h at 4 °C. Aminoacylation percentage was obtained by quantifying the per-lane normalized band intensity in the ImageQuant TL software. The amino acids that displayed sufficient aminoacylated versus non-aminoacylated gel band resolution were subjected to the next step.
2. The remaining aminoacylation reaction was immediately diluted tenfold in the chemical ligation buffer in three separate replicates. The ligation reaction contained 1 μM of the aminoacylated RNA, 4 μM of the 5′-phosphorimidazolide activated hairpin adapter (Supplementary Table 2), 10 μM of the acceptor stem mimic complement (Supplementary Table 2), 200 mM HEPES pH 6.5, 5 mM MgCl$_2$, and 100 mM of HEI pH 6.5. After 90 min at room temperature, 1 μL aliquots were diluted in 9 μL of acidic quenching buffer, and analyzed by standard 20% denaturing urea-PAGE. The efficiency of the ligation reaction was obtained by quantifying the per-lane normalized band intensity in the ImageQuant TL software. The normalized ligation efficiency was obtained by dividing the fraction ligated by the fraction aminoacylated and multiplying by 100%.

### Yeast strains and growth conditions
Yeast strains used in this study are listed in Supplementary Table 1. For chemical ligation validation by acid northern (Fig. 1B), a single colony of S288C was inoculated into YEP glucose (yeast extract, peptone, 2% glucose) and incubated at 30 °C overnight with rotation before dilution to an OD$_{660}$ of 0.2 in YEPD media. The culture was grown to log

phase shaking at 30 °C before a pellet was collected, flash frozen in liquid nitrogen, and stored at −80 °C.

For validation of chemical charging northerns (Fig. 1C), a single colony of WY798 was inoculated into synthetic complete media and incubated at 30 °C overnight with rotation. This culture was diluted to 50 mL of synthetic complete media the next day and allowed to shake overnight at 30 °C before dilution to an $OD_{660}$ of 0.2 in 100 mL synthetic complete media. The culture was grown to log phase at 30 °C, collected by centrifugation, and resuspended in 100 mL room temperature synthetic complete media. The culture was allowed to grow in the new media for 15 min, shaking at 30 °C before it was pelleted, washed with water, flash frozen in liquid nitrogen, and stored at −80 °C.

For the nutrient stress experiment (Fig. 3B–D), 3 colonies of WY795 were inoculated into synthetic complete media and incubated at 30 °C overnight with rotation before dilution to an $OD_{660}$ of 0.2 in 50 mL the same media. The cultures were grown to log phase shaking at 30 °C before pellets were collected by centrifugation. Each pellet was washed by resuspension in a small volume of synthetic media lacking uracil, tryptophan, histidine, and leucine and split into two tubes. Cells were pelleted again and the wash media was removed. One pellet from each original culture was resuspended in 25 mL 30 °C synthetic complete media and the other in 25 mL 30 °C synthetic media lacking leucine. They were allowed to grow in the new media for 15 min, shaking at 30 °C, before they were pelleted, washed with water, flash frozen in liquid nitrogen, and stored at −80 °C.

For the temperature stress experiment (Fig. 3E–H), 3 colonies of JMW 009 and 3 colonies of JMW 510 were inoculated into YEPD media and incubated at 30 °C overnight with rotation before dilution to an $OD_{660}$ of 0.2 in 60 mL YEPD media. The cultures were grown to log phase shaking at 30 °C. At this point, 50 mL of culture was pelleted by centrifugation, washed with water, flash frozen in liquid nitrogen, and stored at −80 °C. 40 mL of 40 °C YP glucose media was added to the remaining 10 mL of culture and these cells were grown for 3 h at 37 °C. They were then pelleted, washed with water, flash frozen in liquid nitrogen, and stored at −80 °C.

### Isolation and chemical ligation of aminoacylated tRNAs from budding yeast

**A complete protocol for aa-tRNA-seq is available on Benchling.** Yeast pellets were thawed on ice and resuspended in 400 μL of cold AES (10 mM NaOAc pH 4.5, 1 mM ETDA pH 8, 0.5% SDS). 400 μL of cold 25:24:1 acid phenol:chloroform:isoamyl alcohol was added. Samples were vortexed for 15 s and allowed to rest on ice for 20 min, vortexing every 5 min. They were then spun at $18,000 \times g$ for 10 min at 40 °C and the aqueous phase was moved to a new tube.

A 0.4× volume of Ampure XP beads (Fisher Scientific A63881) were added to 100 μL of aqueous phase. They were rotated for 2 min at RT and placed on a magnet until the beads had settled. The supernatant was moved to a new tube and quantified via nanodrop. Small RNAs were isolated from 100 μg of this supernatant using a Zymo Research RNA Clean and Concentrator kit (R1018) according to the manufacturer's instructions. Dilution of the bead supernatant for the first step of the kit was done with 10 mM NaOAc pH 4.5, not with water. Small RNA was eluted in 30 μL of 10 mM NaOAc pH 4.5 and quantified via nanodrop. It was stored at −80 °C.

Two 3′ DNA-RNA hybrid splint adapters were designed with different internal sequences, one to ligate to deacylated tRNAs ("uncharged 3′ adapter") and the other to acylated tRNAs ("charged 3′ adapter", see Supplementary Table 2). A universal 5′ adapter was designed to pair with either of the 3′ adapters. Syntheses of these adapters were ordered from IDT and resuspended in water to a concentration of 2 mM. Adapters were run on a 1.5 mM 6% TBU (Tris, boric acid EDTA) V16 polyacrylamide gel with 10 nmol loaded per lane (10-well comb). Staining was not performed and UV shadowing was used to excise the adapters. Gel slices were cut into small pieces and rotated end over end in crush + soak buffer (300 mM NaOAc pH 5.5, 1 mM EDTA pH 8.0, 0.1% SDS) overnight at 4 °C. Nucleic acids were precipitated with 100% ethanol and resuspended in water to a final concentration of 100 μM.

Gel-purified charged 3′ adapter was incubated with a 500-fold molar excess of both imidazole and EDC (1-Ethyl-3-(3-dimethylaminopropyl)carbodiimide) for 2 h at 25 °C to imidazolate the 5′ end of the adapter. 30 μL of cold 99.5%+ acetone saturated with perchlorate and 1 mL of cold 99.5%+ acetone were added to precipitate the imidazolated adapter. The sample was incubated for 20 min on dry ice and then centrifuged at maximum speed for 10 min at 4 °C. The supernatant was removed and the pellet was washed twice with 1 mL 1:1 acetone: diethyl ether. The pellet was dried in a speed vacuum and resuspended in 10 mM imidazole pH 7.0 to a final concentration of 200 μM.

Small RNA (15-50 pmol) was incubated in 100 mM MES pH 5.5, 2.5 mM $MgCl_2$, a fivefold molar excess of both imidazolated 3′ adapter and gel-purified 5′ adapter, and 50 mM HEI pH 6.5 for 30 min at 25 °C, establishing a phosphoramidate covalent linkage between the 3′ splint adapter and aminoacylated tRNAs. Ligated products were purified by crush and soak (0.3 M NaOAc pH 5.5, 1 mM EDTA pH 8.0, 0.1% SDS) at 4 °C, overnight) from a 10% TBU polyacrylamide gel, isolating the regions between 70 and 150 nts. The eluate was precipitated by addition of ethanol and GlycoBlue coprecipitant (Invitrogen) resuspended in a small volume of 10 mM NaOAc pH 4.5 and quantified via absorbance at 260 nm (Nanodrop).

### Northern blotting

Transfer RNA charging was measured by acidic northern blot, resolving 75 ng of small RNA on a gel (6% 19:1 acrylamide, 0.1 M sodium acetate, pH 4.5, 8 M urea) 42 cm in length which was run at 450 V for 22 h in a cold room. For chemical-charging northern blots, chemically ligated tRNA (220 ng) was loaded onto 10% TBU polyacrylamide gels (7.5 cm, 6% 19:1 acrylamide, 1× TBE, 8 M urea) and electrophoresed in 1× TBE at room temperature at 250 V for 40 min. Acid-urea and TBU gels were transferred to charged nylon membranes (Hybond N+, GE) via electroblot transfer at 1 Amp for 1 h for acid gels and 3 mA/cm² based on the membrane area for 35 min for TBU gels. After transfer, membranes were UV-crosslinked at 254 nm using a 120 mJ dose and blocked in ULTRAhyb-Oligo (Thermo) before an incubation with [32]P-labeled oligonucleotide probes in ULTRAhyb-Oligo overnight at 42 °C (Supplementary Table 2). Membranes were washed four times at 42 °C (2× SSC, 0.1% SDS), wrapped in plastic, and exposed to a phosphor-imager screen before imaging on a Typhoon 9400 (GE Healthcare). Membranes were stripped with two 30 min washes in 2% SDS at 80 °C, prior to reblocking and incubation with labeled probe.

### Nanopore library generation for budding yeast tRNAs

Gel-purified tRNAs from chemical ligation were enzymatically ligated to capture deacylated tRNAs with 3′ splint adapters and attach 5′ adapters to all tRNA. tRNA from the first ligation (20 pmol) was incubated in a 20 μL reaction consisting of 10% PEG 8000, 1 μL of RNase inhibitor (Watchmaker Genomics), 9 pmol gel-purified uncharged 3′ splint adapter, 9 pmol gel-purified 5′ adapter, 1× T4 RNA ligase 2 buffer, and 2 μL of T4 RNA ligase 2 (homemade preparation, 0.74 mg/mL). This ligation was incubated at 25 °C for 30 min.

Ligation products were purified by addition of a 1.8× volume of tRNA beads (BioDynami), mixing by pipetting, and incubation on ice for 4 min, followed by magnetic separation. The supernatant was discarded. Beads were washed with 180 μL 80% EtOH and air dried. Beads were resuspended in 13 μL of water, and the elution was moved to a new tube and quantified.

Splint-adapter-ligated tRNAs are next ligated to RT adapters (RTA) (provided in the RNA004 ONT kit): 12.5 μL of sample was incubated with 1.5 μL RTA, 0.5 μL RNase inhibitor (Watchmaker Genomics), 4 μL T4 DNA ligase buffer, and 1.5 μL T4 DNA ligase (Watchmaker Genomics, 600 U/μL) for 30 min at 25 °C, and cleaned up at RT using the tRNA beads as above, using a 1.35× volume of beads, and elution in 26 μL water. Each sample was quantified with the Quant-iT Qubit dsDNA HS kit and the library size distribution was confirmed by Agilent TapeStation (HS DNA 1000).

Finally, ligation products were ligated to ONT's RNA ligation adapter (RLA) on the same day that sequencing was conducted. 50–400 fmol of sample in 23 μL was incubated with 6 μL RLA (ONT RNA004), 8 μL T4 DNA ligase buffer, and 3 μL T4 DNA ligase (Watchmaker Genomics, 600 U/μL) for 30 min at 25 °C. These final ligations were cleaned up using a 1.8× volume of Ampure XP SPRI beads (Beckman Coulter), and washed with WSB wash buffer (ONT) following the protocol for ONT SQK-RNA004.

### Sequencing run conditions for budding yeast tRNAs
Libraries were loaded onto "RNA" flow cells on a PromethION P2 Solo instrument connected to a A5000 GPU workstation or a PromethION P2integrated instrument, using MinKNOW software version 24.06.10. We found the throughput of aa-tRNA-seq to be comparable or superior to previous nanopore tRNA sequencing approaches[32,86], collecting a median >8 million reads for biological tRNA sequencing libraries, and median ~250 thousand reads for synthetic tRNA sequencing libraries.

### Base-calling and alignment
Libraries were basecalled with Dorado v0.7.2 (ONT, https://github.com/nanoporetech/dorado) using the "super high accuracy" (rna004_130bps_sup) v5.0.0 model and –emit-moves parameter. Basecalled bams were converted to fastq format using samtools v. 1.21[87] with -T "*" flag to retain move tables, and then aligned to using BWA-MEM version 0.7.16-r1181[88] with the parameters `bwa mem −C −W 13 −k 6 −x ont2d`, enabling transfer of move tables to aligned bams.

To evaluate the performance of an alignment-based classification approach for identifying charged vs. uncharged tRNA reads, aligned bams were further filtered to contain reads mapped to full length tRNAs. Aside from this filtering, no data were excluded from the analysis. tRNA reference files were constructed by appending CCA sequences to each mature tRNA sequence, along with the unique 3′ and universal 5′ adapter sequences, and primary alignments for each read assessed. A Snakemake[89] analysis pipeline is available at https://github.com/rnabioco/aa-tRNA-seq-pipeline.

### Remora model training and validation
To train a machine learning model distinguishing charged and uncharged tRNAs we used the Remora software (v3.2, https://github.com/nanoporetech/remora) and training procedure for modified nucleosides. First, we prepared fully charged yeast tRNA libraries (treated as modified base in the training) and deacetylated libraries (treated as modified base control). To make our model universal for all tRNAs we defined a 6-nt modification kmer spanning the universal CCA 3′ end of tRNA and the first three nucleotides of the 3′ adapter (CCAGGC), where the underlined G was defined as the modification site. We extract chunks from both libraries using `remora dataset prepare` with the default remora 9mer table for rescaling and following parameters: `−refine-rough-rescale −reverse-signal −motif CCAGGC` 3. We prepared training configuration files with `Remora dataset make_config` using `−dataset-weights 1 1`. Finally, we trained the model using the parameters: `−model ConvLSTM_w_ref.py −chunk-context 200 200 −num-test-chunks 20000`. After internal remora validation on 20000 chunks, this model was validated on independent libraries using reference-anchored remora inference (`remora infer from_pod5_and_bam`

`−reference-anchored`). We used ML tags from inference output bams to calculate model performance, with the following assumption: ML < 200 = uncharged tRNA, ML ≥ 200 = charged tRNA.

Pairwise machine learning classifiers to distinguish individual amino acids (binary recognition between pairs of amino acids totaling 380 models) were trained using Remora v. 3.2 using a procedure similar to that described above, using Flexizyme-charged synthetic tRNA reads for the training. Synthetic tRNA was aligned to a reference sequence with one nucleotide ("T") inserted between the CCA sequence at the 3′ terminus and the start of the 3′ adapter sequence. Pairwise models were trained with one amino acid (i) treated as modified base and the second one (j) as modified base control on the CCAT motif, with the inserted T identified as the modification position for Remora training; the mean F1 score for AA(i,j) and AA(j,i) are displayed in the performance data in Fig. 4J. 10000 chunks were used for internal Remora validation, except for pairwise comparisons with alanine, where 9000 chunks were used due to lower library depth for the alanine-charged library. F1 scores were calculated for all models using the equation: $2TP/(2TP + FP + FN)$.

### Nanopore signal analysis
Signal metrics (dwell time and trimmed means of ionic current) at reference-anchored positions were extracted from POD5 files using the Remora API and stored in TSV files for analysis and plotting in R. Example scripts for signal extraction and plotting will be available at https://github.com/rnabioco/aa-tRNA-seq.

### Statistics & reproducibility
Sample sizes for synthetic tRNA ligation and nanopore sequencing were selected based on established practices in RNA biochemistry and pilot experiments that demonstrated clear signal resolution. For biological tRNA libraries, sample sizes were guided by the practical throughput of Oxford Nanopore direct RNA sequencing, which improved over the course of the study but remained a limiting factor due to singleplex flow cell use. For nutrient limitation and temperature stress experiments, three biological replicates were used to capture biological variability and support reproducibility of observed effects. These sample sizes are consistent with prior work using northern blotting and nanopore sequencing to assess tRNA charging and abundance.

Experiments were not randomized, and investigators were not blinded to sample allocation or outcome assessment.

Statistical comparisons of tRNA abundance and aminoacylation levels were performed using two-sided Z tests on normalized read counts, with Benjamini–Hochberg correction applied for multiple hypothesis testing. Changes in gel band intensity were quantified using ImageQuant TL software, and values were normalized to total lane signal. For classification model performance (e.g., Remora models), F1 scores were computed as $2TP/(2TP + FP + FN)$ using held-out test sets. All statistical analyses were conducted in Python or R.

### Reporting summary
Further information on research design is available in the Nature Portfolio Reporting Summary linked to this article.

## Data availability
Raw nanopore sequencing data generated in this study has been deposited as POD5 files in the European Nucleotide Archive under accession number PRJEB90828. Specific links to access the ENA POD5 files are available at https://github.com/rnabioco/aa-tRNA-seq/blob/main/README.md. Per-read current and dwell time metrics for synthetic tRNA libraries are available on Zenodo under https://doi.org/10.5281/zenodo.14194755 [https://doi.org/10.5281/zenodo.14194755]. Source data supporting the findings of this study are provided with this paper. Source data are provided with this paper.

## Code availability

All code used for analysis and figure generation, as well as a reproducible pipeline for aa-tRNA-seq data processing, is available at the associated GitHub repository: https://github.com/rnabioco/aa-tRNA-seq. A permanent archive of the version used in this study is available on Zenodo under https://doi.org/10.5281/zenodo.15653411 [https://doi.org/10.5281/zenodo.15653411][90].

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

## Acknowledgements

This work was supported by the National Institutes of Health (R35 GM119550 to J.H.) and National Science Foundation (Award #2330283 to J.H.). J.W.S. is an Investigator of the HHMI. We thank Filip Bošković for inspiring discussions during the project planning phase, Ron Wek and Eric Phizicky for yeast strains, Marcus Stoiber for advice and assistance with Remora model training, and members of the Hesselberth and Szostak labs for constructive feedback.

## Author contributions

L.W., A.R., J.S., and J.H. designed and conceived the experiments. L.W., A.R., M.S., K.D., K.R., and S.P. performed experiments. L.W., A.R., J.S., and J.H. wrote the paper.

## Competing interests

A.R., L.K.W., J.W.S., and J.R.H have submitted a patent application to the USPTO (PCT/US2025/031145) pertaining to the chemistry, sequencing method, and machine learning aspects of this work. L.K.W. has received travel and accommodation expenses from Oxford Nanopore Technologies to present at scientific meetings, and has participated in beta tests of ONT's direct RNA sequencing products. L.K.W. is a co-founder of an early-stage entity that may seek to license and commercialize aspects of the technology described in this work. The remaining authors declare no competing interests.
