## [Transparent Peer Review file · Nature Communications]

Nanopore sequencing of intact aminoacylated tRNAs

Corresponding Author: Dr Jay Hesselberth

Version 0:

Reviewer comments:

Reviewer #1

(Remarks to the Author)

White, Radakovic et al. describe “aa-tRNA-seq”, a method that enables sequencing of both the tRNA and its associated amino acid using nanopore sequencing. Using this method, charged and uncharged tRNA can be differentiated. Employing machine learning, they show the amino acids of the aminoacyl-tRNA display signal-based features that allow them to be distinguished from each other.

Sequencing and distinguishing the amino acid of aminoacyl-tRNA has not previously been demonstrated. It is a significant innovation certain to be of high interest to readers. It opens up new ways of studying the regulation of, and errors in tRNA aminoacylation.

The crucial technical innovation in this paper is the chemical ligation of the tRNA-associated amino acid to a short oligonucleotide. This is followed by enzymatic ligations to make sequence-ready aminoacylated tRNA. Without the chemical ligation linking the adapter to the amino acid, sensing using nanopore direct RNA sequencing could not work because it requires the presence of nucleic acid adapters at the 3' end of the molecule. Getting the system to work with all 20 amino acids was a challenging undertaking. The authors found tRNA charged with certain amino acids become stuck and ejected while sequencing. They were able to overcome this issue by introducing DNA residues into the 5' adapter strand.

This paper is well written, thorough, and its assertions and methodology are well documented. Sequencing experiments were run with accompanying acid northern blots as controls. The authors demonstrate the ability of aa-tRNA-seq to measure charging and relative aminoacyl-tRNA abundance by verifying previous findings of yeast mutations known to impact abundance, stability and charging of specific tRNA isodecoders and make some additional observations not previously reported.

This paper clearly merits publication. However, clarifications to the writing and some additions to the discussion section are requested to make this manuscript ready for publication.

Additional discussion requested:

While the ability to study and measure tRNA aminoacylation is demonstrated, this work also lays groundwork and can provide insights for a future method for sequencing the tRNA, including its all modifications and amino acid simultaneously. The final paragraph of the discussion addresses this.

There are three further question related to this:

Does the chemical ligation procedure alter specific RNA modifications?

Will running deacylated and aminoacylated libraries together give a more comprehensive representation of tRNA species?

Will it be important to address the increase in base calling errors seen at the 3' end of the tRNA near the amino acid?

Clarifications:

1. The first paragraph of the Results has inadequate explanation of preliminary studies (Fig S1A)

The authors state:

'we tested and found that a synthetic tRNA-Gly-GCC aminoacylated with glycine or lysine using the Flexizyme underwent chemical ligation with moderate kinetics, while the non-aminoacylated tRNA yielded no detectable product (Fig. S1A).'

Fig. S1A is unclear because the lower 'tRNA' band in the gel contains both tRNA that is and is not aminoacylated. But, these can't be differentiated on that particular gel. This could be confusing for someone outside the field, although it is somewhat explained later in the manuscript.

2. Please review the first paragraph of the Results. References to Fig S1B and S1C are switched in the text.

3. Figure 2A legend clarification

It reads:

(A) Schematic illustrating ground truth libraries for model training. Library (i) was prepared by only chemical ligation to biological aminoacylated tRNAs. Library (ii) was prepared by only enzymatic ligation to deacylated biological tRNAs.

I believe "only" this refers only to the ligation on the 3' end of the molecule. Both have enzymatic ligation to the 5' end, correct? This is clear in the text, but should also be noted in the figure legend.

4. Clarification regarding abundance measurements from northern blots

Question: When relative abundance levels are measured on northern blots, for example in the Fig 3B legend (and other figure legends) it says "relative abundances represent within-replicate normalized levels of total tRNA". Is this determined by measuring the sum of the densities of the tRNA (lower) and the aminoacylated tRNA (upper) band for the control vs experimental lanes? Or by total tRNA do you mean all the tRNAs in the sample including those not probed for?

5. Methods

Add the concentrations/units for the reagents from Watchmaker Genomics.

6. Fig S7A clarity

Legend reads:

Fig S7 (A) Log₂ fold change in tRNA abundance and tRNA charging percent charged reads in the RTD-sensitive budding yeast strain *tan1Δ trm44Δ* and the RTD-resistant strain *tan1Δ trm44Δ met22Δ* after 3 hours growth at the nonpermissive or permissive temperatures.

Suggest that "charged reads" be in parentheses.

7. Ref. 51 has been published in PLOS, please amend citation information.

(Remarks on code availability)

Reviewer #2

(Remarks to the Author)

The authors describe a novel approach that permits nanopore sequencing of both charged and uncharged tRNAs in the same run. This is accomplished by subjecting the tRNA to two sequential ligation reactions. In the first step charged tRNAs are chemically ligated to a 3'-RNA adaptor which contains a 5'-phosphorimidazolide group. This group forms a covalent link to the alpha-amino group of a charged tRNA in the presence of the catalyst 1-(2-hydroxyethyl)imidazole and an RNA splint. Next, uncharged tRNAs are enzymatically ligated to a 3'-RNA adaptor that contains a 5'-phosphate group using T4 RNL2. At the same time the RNA splint is ligated to the 5'-terminus of both charged and uncharged tRNAs. Following addition of the standard Oxford nanopore adaptors to the 3'-end of each tRNA, nanopore sequencing is performed.

Efficiency of the chemical ligation step with respect to each isoacceptor family was evaluated by reacting yeast tRNA (either charged or deacylated) with the phosphorimidazolated 3'-adaptor followed by analysis on an acid PAGE gel. Uncharged, charged, and chemically ligated charged tRNAs can be readily resolved and quantified in such a gel. Each isoacceptor was visualized by Northern hybridization thus requiring multiple cycles of stripping and hybridization. For 5 of the isoacceptor families the chemical ligation reaction was quantitative and for another 10 families the efficiency was 60-80%. Technical issues prevented analysis of the remaining families.

Given that the efficiency of the chemical ligation is dependent upon the amino acid, absolute comparisons of charging between isoacceptor families cannot be made thus limiting analyses to relative differences. Since the activated 3'-adaptor reacts with amino groups, it is curious that the authors did not comment on whether other naturally occurring amino groups on tRNA, such as lysidine or acp3U, are also substrates. This should be investigated.

Several years ago, Suga demonstrated that charged tRNAs could be separated from uncharged tRNAs in a neutral PAGE gel by a two-step reaction sequence using NHS-biotin and streptavidin. Unlike the chemical ligation reaction described here, the Suga protocol uses a very high molar excess of NHS-biotin to tRNA which results in efficient coupling to any amino acid. Since the authors demonstrate that their chemical ligation reaction can also be monitored in a neutral PAGE gel, the two strategies should be compared in the manuscript.

It is convincingly demonstrated that the nanopore signals can readily differentiate between charged and uncharged tRNAs that have been processed by the two-step ligation protocol. Unfortunately, at the present time the identity of the amino acid cannot be assigned. This is unfortunate since the ability to do so would allow mischarging to be monitored on a global basis. If this limitation could be overcome, aa-tRNA-seq would have widespread and important applications.

A major concern of this reviewer is the absence of any discussion on why cDNA synthesis was not carried out prior to nanopore sequencing of the tRNA. It is well known that converting RNA to an RNA-DNA hybrid not only stabilizes the RNA prior to nanopore analysis but also presents a uniform hybrid to the helicase motor protein, both of which improve the performance of RNA nanopore sequencing. Omission of a cDNA synthesis step is likely because no reverse transcriptase would be able to bypass an amino acid. This issue needs to be fully discussed in the manuscript.

(Remarks on code availability)

Version 1:

Reviewer comments:

Reviewer #1

(Remarks to the Author)

Revision 1: The authors have fully addressed my previous questions and have provided the requested clarifications to the manuscript. I recommend the paper for publication.

(Remarks on code availability)

Reviewer #2

(Remarks to the Author)

The authors have satisfactorily addressed all my concerns and modified the manuscript accordingly. I recommend publication in Nature Communications.

(Remarks on code availability)

REVIEWER COMMENTS for NCOMMS-25-15813-T

Reviewer #1 (Remarks to the Author):

White, Radakovic et al. describe “aa-tRNA-seq”, a method that enables sequencing of both the tRNA and its associated amino acid using nanopore sequencing. Using this method, charged and uncharged tRNA can be differentiated. Employing machine learning, they show the amino acids of the aminoacyl-tRNA display signal-based features that allow them to be distinguished from each other.

Sequencing and distinguishing the amino acid of aminoacyl-tRNA has not previously been demonstrated. It is a significant innovation certain to be of high interest to readers. It opens up new ways of studying the regulation of, and errors in tRNA aminoacylation.

The crucial technical innovation in this paper is the chemical ligation of the tRNA-associated amino acid to a short oligonucleotide. This is followed by enzymatic ligations to make sequence-ready aminoacylated tRNA. Without the chemical ligation linking the adapter to the amino acid, sensing using nanopore direct RNA sequencing could not work because it requires the presence of nucleic acid adapters at the 3' end of the molecule. Getting the system to work with all 20 amino acids was a challenging undertaking. The authors found tRNA charged with certain amino acids become stuck and ejected while sequencing. They were able to overcome this issue by introducing DNA residues into the 5' adapter strand.

This paper is well written, thorough, and its assertions and methodology are well documented. Sequencing experiments were run with accompanying acid northern blots as controls. The authors demonstrate the ability of aa-tRNA-seq to measure charging and relative aminoacyl-tRNA abundance by verifying previous findings of yeast mutations known to impact abundance, stability and charging of specific tRNA isodecoders and make some additional observations not previously reported.

This paper clearly merits publication. However, clarifications to the writing and some additions to the discussion section are requested to make this manuscript ready for publication.

Additional discussion requested:

While the ability to study and measure tRNA aminoacylation is demonstrated, this work also lays groundwork and can provide insights for a future method for sequencing the tRNA, including its all modifications and amino acid simultaneously. The final paragraph of the discussion addresses this.

There are three further question related to this:

Does the chemical ligation procedure alter specific RNA modifications?

- The chemical ligation positions an activated 5'-phosphate near the alpha-amino group of a charged amino acid through splinting of the CCA tRNA overhang. In the absence of this favorable steric arrangement, we do not anticipate any internal modifications will react with the ImP moiety.
- Consistent with the above, in conditions of complete chemical deacylation (Fig. 1B), only trace amounts (<0.1%) of tRNA reacted with an EDC-activated adapter as visualized on acid northern blot, suggesting that any non-amino acid reactivity (be that from the diol or other nucleobase groups), is minimal.

- In yeast, there are no documented modifications containing a primary amine [REF MODOMICS].
- For organisms that do contain modifications with primary amine nucleophiles (e.g., lysidine in bacteria), these occur in other regions of the tRNA outside of the splinted region, so we would not expect them to react.
- In terms of the data in this manuscript, we did perform a preliminary analysis comparing base calling error signatures for the same isodecoder between charged and uncharged molecules to look for changes in modification that depend on aminoacylation and find a handful of potential sites, but currently cannot distinguish between changes in biological modification and ligation-associated modification (though we don't expect this).

Will running deacylated and aminoacylated libraries together give a more comprehensive representation of tRNA species?

We are unclear precisely what the reviewer is asking with this question, so will address a few possible interpretations below:

- The present manuscript describes the initial generation of distinct budding yeast libraries containing (i) chemically deacylated and (ii) untreated (representing a biological mix of charged & uncharged tRNA) libraries for the purpose of training a model to distinguish charged from uncharged tRNAs. However, after training and validation of this model (as illustrated in Fig. 2), all biological samples are sequenced as a mixed pool of charged and uncharged tRNAs captured via chemical and enzymatic ligation respectively, in order to estimate the percent of each tRNA isodecoder that is charged vs uncharged. We have made some slight edits to wording in the Results section to further clarify this point.
- The underrepresentation of charged tRNA molecules due to systematic sequencing bias (Fig. 3A / Fig. S4) does raise the question regarding how comprehensive a representation the current method provides of biological tRNA repertoires. We plan to address this in future experiments via the addition of acylated and deacylated tRNA spike ins at known concentrations to better normalize for this source of bias.
- Another possibility to address the bias discussed above would be to compare the outputs of aa-tRNA-seq libraries to chemically deacylated samples worked up entirely via enzymatic ligation. We think this could provide some informative comparisons at the level of isodecoder abundance and tRNA modification; however, care would be required as amino acids are differentially labile, and therefore incomplete chemical deacylation could also bias the pool in a different direction.

Will it be important to address the increase in base calling errors seen at the 3' end of the tRNA near the amino acid?

We do not believe basecalling errors are a major issue for aa-tRNA-seq because the signal distortions generated by amino acids are predominantly restricted to a window spanning the CCA terminus and 3' RNA adapter. However, these distortions (and resulting errors generated by the Dorado basecaller) did impede the performance of our initial alignment-based strategy to differentiate charged from uncharged molecules based on 3' adapter sequence alone, as described in the text and panel for Fig. 2F. This sequence alignment-based classification approach is far more sensitive to basecalling errors than the signal-based ML model which we selected for ongoing use.

Since the 3' end of tRNAs has invariant sequence and lacks annotated modifications, we further do not expect

amino acid derived basecalling errors within this region to have any downstream impacts on (a) alignment to the appropriate tRNA isodecoder or (b) modification detection.

Clarifications:

1. The first paragraph of the Results has inadequate explanation of preliminary studies (Fig S1A)

The authors state:

‘we tested and found that a synthetic tRNA-Gly-GCC aminoacylated with glycine or lysine using the Flexizyme underwent chemical ligation with moderate kinetics, while the non-aminoacylated tRNA yielded no detectable product (Fig. S1A).’

Fig. S1A is unclear because the lower ‘tRNA’ band in the gel contains both tRNA that is and is not aminoacylated. But, these can’t be differentiated on that particular gel. This could be confusing for someone outside the field, although it is somewhat explained later in the manuscript.

We have added an additional sentence to the figure legend to make the identity of this band clearer.

2. Please review the first paragraph of the Results. References to Fig S1B and S1C are switched in the text.

Edits made.

3. Figure 2A legend clarification

It reads:

(A) Schematic illustrating ground truth libraries for model training. Library (i) was prepared by only chemical ligation to biological aminoacylated tRNAs. Library (ii) was prepared by only enzymatic ligation to deacylated biological tRNAs.

I believe “only” this refers only to the ligation on the 3’ end of the molecule. Both have enzymatic ligation to the 5’ end, correct? This is clear in the text, but should also be noted in the figure legend.

That is correct. To clarify for the reader, we have changed this figure legend to read “ Library (i) was prepared by only chemical ligation of the 3’ adapter to biological aminoacylated tRNAs, followed by gel purification and enzymatic ligation of the 5’ adapter. Library (ii) was prepared by enzymatic ligation of both adapters to deacylated biological tRNAs.”

4. Clarification regarding abundance measurements from northern blots

Question: When relative abundance levels are measured on northern blots, for example in the Fig 3B legend (and other figure legends) it says “relative abundances represent within-replicate normalized levels of total tRNA”. Is this determined by measuring the sum of the densities of the tRNA (lower) and the aminoacylated tRNA (upper) band for the control vs experimental lanes? Or by total tRNA do you mean all the tRNAs in the sample including those not probed for?

This is determined for the probed tRNA only using the sum of the densities per lane. We thank the reviewer for pointing out that our figure legend could have been more clear here and have rephrased the respected sentences to be less ambiguous.

5. Methods

Add the concentrations/units for the reagents from Watchmaker Genomics.

Thank you, this is now clarified as the 600 U/ μ L product in the methods.

6. Fig S7A clarity

Legend reads:

Fig S7 (A) Log₂ fold change in tRNA abundance and tRNA charging percent charged reads in the RTD-sensitive budding yeast strain *tan1 Δ trm44 Δ* and the RTD-resistant strain *tan1 Δ trm44 Δ met22 Δ* after 3 hours growth at the nonpermissive or permissive temperatures.

Suggest that “charged reads” be in parentheses.

Clarifying change made.

7. Ref. 51 has been published in PLOS, please amend citation information.

Updated accordingly (NB: due to a few other changes this is no longer ref #51).

Reviewer #2 (Remarks to the Author):

The authors describe a novel approach that permits nanopore sequencing of both charged and uncharged tRNAs in the same run. This is accomplished by subjecting the tRNA to two sequential ligation reactions. In the first step charged tRNAs are chemically ligated to a 3'-RNA adaptor which contains a 5'-phosphorimidazole group. This group forms a covalent link to the alpha-amino group of a charged tRNA in the presence of the catalyst 1-(2-hydroxyethyl)imidazole and an RNA splint. Next, uncharged tRNAs are enzymatically ligated to a 3'-RNA adaptor that contains a 5'-phosphate group using T4 RNL2. At the same time the RNA splint is ligated to the 5'-terminus of both charged and uncharged tRNAs. Following addition of the standard Oxford nanopore adaptors to the 3'-end of each tRNA, nanopore sequencing is performed.

Efficiency of the chemical ligation step with respect to each isoacceptor family was evaluated by reacting yeast tRNA (either charged or deacylated) with the phosphorimidazolated 3'-adaptor followed by analysis on an acid PAGE gel. Uncharged, charged, and chemically ligated charged tRNAs can be readily resolved and quantified in such a gel. Each isoacceptor was visualized by Northern hybridization thus requiring multiple cycles of stripping and hybridization. For 5 of the isoacceptor families the chemical ligation reaction was quantitative and for another 10 families the efficiency was 60-80%. Technical issues prevented analysis of the remaining families.

Given that the efficiency of the chemical ligation is dependent upon the amino acid, absolute comparisons of charging between isoacceptor families cannot be made thus limiting analyses to relative differences. Since the activated 3'-adaptor reacts with amino groups, it is curious that the authors did not comment on whether other

naturally occurring amino groups on tRNA, such as lysidine or acp3U, are also substrates. This should be investigated.

As discussed above in response to the first reviewer, the conditions of complete chemical deacylation in Fig. 1B show low reactivity of deacylated tRNAs with EDC-activated adapter as visualized on acid northern blot, consistent with minimal non-amino acid reactivity. We also note that no modifications containing primary amines are annotated in MODOMICS for *S. cerevisiae* tRNAs. However, we agree with the reviewer that it may be interesting to explore the background reactivity of tRNAs from various different organisms with different repertoires of tRNA modifications in the future.

Several years ago, Suga demonstrated that charged tRNAs could be separated from uncharged tRNAs in a neutral PAGE gel by a two-step reaction sequence using NHS-biotin and streptavidin. Unlike the chemical ligation reaction described here, the Suga protocol uses a very high molar excess of NHS-biotin to tRNA which results in efficient coupling to any amino acid. Since the authors demonstrate that their chemical ligation reaction can also be monitored in a neutral PAGE gel, the two strategies should be compared in the manuscript.

We thank the reviewer for drawing this inadvertent omission to our attention. We have added the following text to the end of the section titled “A chemical-northern blot simplifies analysis of aminoacylated tRNA” and also referenced the previous works in the Discussion section:

This strategy of stabilizing the aminoacyl ester and appending a bulky group to the alpha-amine to make it amenable to standard PAGE is reminiscent of earlier studies that leveraged highly-reactive biotin-NHS reagents to biotinyrate the alpha-amine, which after incubation with streptavidin could be resolved from deacylated tRNA with standard PAGE [citations to PMID: 9162902, PMID: 14652075, and PMID: 33036365]. However, our approach offers two key advantages: (1) very low background reactivity with deacylated tRNA (Figure 1B) due to the low concentration of the activated adapter used in the reaction and (2) direct modification of aminoacylated tRNA with an oligonucleotide adapter, which allows both the resolution of the aminoacylated from the deacylated tRNA by standard PAGE and the subsequent preparation of aminoacylated tRNA libraries for nanopore sequencing (*vide infra*).

It is convincingly demonstrated that the nanopore signals can readily differentiate between charged and uncharged tRNAs that have been processed by the two-step ligation protocol. Unfortunately, at the present time the identity of the amino acid cannot be assigned. This is unfortunate since the ability to do so would allow mischarging to be monitored on a global basis. If this limitation could be overcome, aa-tRNA-seq would have widespread and important applications.

The 380 pairwise models trained and evaluated in Fig 4I-J represent a first effort towards discrimination of amino acid identity, and are on par with the present machine learning model accuracy for discriminating amino acids in nanopore peptide sequencing approaches (see PMID 39261738 - Motone *et al.* 2024). However, we agree with the reviewer’s assessment that the current models have substantial headroom for improvement, and are focusing future efforts on refining them to achieve higher accuracy for application to biological samples.

Nevertheless, we note that for many questions in tRNA biology comprehensive *de novo* classification of all 20 canonical amino acids is likely unnecessary, as (a) most tRNAs can be presumed to be charged with their

cognate amino acids and (b) tRNA misaminoacylation is not an unbiased event wherein any of the other 19 non-cognate amino acids have an equal probability of being added by a synthetase.

Our future studies will focus on model improvement and evaluation of our ability to measure misaminoacylation, beginning with AARS editing mutants known to generate discrete and predictable misaminoacylation events.

A major concern of this reviewer is the absence of any discussion on why cDNA synthesis was not carried out prior to nanopore sequencing of the tRNA. It is well known that converting RNA to an RNA-DNA hybrid not only stabilizes the RNA prior to nanopore analysis but also presents a uniform hybrid to the helicase motor protein, both of which improve the performance of RNA nanopore sequencing. Omission of a cDNA synthesis step is likely because no reverse transcriptase would be able to bypass an amino acid. This issue needs to be fully discussed in the manuscript.

Omission of a reverse transcription step is not unusual in nanopore tRNA sequencing libraries. cDNA conversion is an optional but recommended step in Oxford Nanopore direct RNA sequencing protocols to increase sequencing yield; however, RT was not included in the foundational proof of concept work that sequenced deacylated *E. coli* tRNAs from Thomas *et al.* 2019. Further, Lucas *et al.* 2024 directly tested the impact of adding reverse transcription to nano-tRNA-seq libraries from budding yeast in the previous (RNA002) ONT chemistry and found it increased throughput by <1.4fold.

In the case of aa-tRNA-seq, we made a deliberate design decision to omit cDNA conversion and have not tested addition of a reverse transcription step to our protocol. We can speculate on several potential issues that might arise during such tests:

- First, the clear question raised by the reviewer of whether any RT could bypass an amino acid
- Second, if the above were possible, this processivity would likely not be the same from one amino acid to another, generating an additional source of bias in the protocol
- Third, while the amino acid bridged tRNA-adaptor construct is stable at room temperature, we have not tested how labile this ester is when incubated at 42°C or higher for reverse transcription

We appreciate the reviewer's feedback on this matter, and have added additional text to the discussion section to clarify this point.